SOFTWARE

# Pairtools: From sequencing data to chromosome contacts

Open2C[1]*, Nezar Abdennur[2,3], Geoffrey Fudenberg[4], Ilya M. Flyamer[5]*, Aleksandra A. Galitsyna[6,7]*, Anton Goloborodko[7]*, Maxim Imakaev[6], Sergey V. Venev[3]

1 https://open2c.github.io/, 2 Program in Bioinformatics and Integrative Biology, University of Massachusetts Chan Medical School, Worcester, Massachusetts, United States of America, 3 Department of Systems Biology, University of Massachusetts Chan Medical School, Worcester, Massachusetts, United States of America, 4 Department of Computational and Quantitative Biology, University of Southern California, Los Angeles, California, United States of America, 5 Friedrich Miescher Institute for Biomedical Research, Basel, Switzerland, 6 Institute for Medical Engineering and Sciences, Massachusetts Institute of Technology (MIT), Cambridge, Massachusetts, United States of America, 7 Institute of Molecular Biotechnology of the Austrian Academy of Sciences (IMBA), Vienna BioCenter (VBC), Vienna, Austria

* open.chromosome.collective@gmail.com; ilia.fliamer@fmi.ch (IMF); galitsyn@mit.edu (AAG); anton.goloborodko@imba.oeaw.ac.at (AG)

**Data Availability Statement:** Open-source code is freely available at https://github.com/open2c/pairtools. Additional documentation and interactive tutorial are available at https://pairtools.

## Abstract

The field of 3D genome organization produces large amounts of sequencing data from Hi-C and a rapidly-expanding set of other chromosome conformation protocols (3C+). Massive and heterogeneous 3C+ data require high-performance and flexible processing of sequenced reads into contact pairs. To meet these challenges, we present *pairtools*–a flexible suite of tools for contact extraction from sequencing data. *Pairtools* provides modular command-line interface (CLI) tools that can be flexibly chained into data processing pipelines. The core operations provided by *pairtools* are parsing of.sam alignments into Hi-C pairs, sorting and removal of PCR duplicates. In addition, *pairtools* provides auxiliary tools for building feature-rich 3C+ pipelines, including contact pair manipulation, filtration, and quality control. Benchmarking *pairtools* against popular 3C+ data pipelines shows advantages of *pairtools* for high-performance and flexible 3C+ analysis. Finally, *pairtools* provides protocol-specific tools for restriction-based protocols, haplotype-resolved contacts, and single-cell Hi-C. The combination of CLI tools and tight integration with Python data analysis libraries makes *pairtools* a versatile foundation for a broad range of 3C+ pipelines.

## Author summary

Our study introduces *pairtools*, a computational suite for extracting pairwise contacts from Hi-C and the rapidly-expanding constellation of chromosome conformation protocols (3C+). These experiments use DNA sequencing to measure the 3D structure of chromosomes inside cells. However, specialized software is needed to extract chromosome contacts from the raw sequencing data. *Pairtools* provides fast, flexible, and modular command-line tools and a Python framework to bridge this gap. We show *pairtools* can process data from many Hi-C protocol variants beyond standard Hi-C and is easily

readthedocs.io/. Pairtools is integrated into the high-performant nextflow-based pipeline distiller: https://github.com/open2c/distiller-nf/. Code for generating manuscript figures available at: https://github.com/open2c/open2c_vignettes/tree/main/pairtools_manuscript/. Benchmarks are available at https://github.com/open2c/pairtools/tree/master/doc/examples/benchmark/.

**Funding:** AAG, NA, and SV acknowledge support from the Center for 3D Structure and Physics of the Genome, funded by the National Institutes of Health Common Fund 4D Nucleome Program UM1-HG011536. SV is additionally supported by R01 HG003143. AAG was partially supported by grant 17-00-00180, IMBA and the Austrian Academy of Sciences (OeAW) during the early development of the project. GF is supported by R35 GM143116-01. Work in the laboratory of AG is supported by the Austrian Academy of Sciences and the Austrian Science Fund (FWF) grant SFB F 8804-B "Meiosis". The funders had no role in study design, data collection and analysis, decision to publish, or preparation of the manuscript.

**Competing interests:** The authors have declared that no competing interests exist.

integrated into pipelines for high-throughput 3D genome data processing. By converting sequence data into tables of chromosome contacts, *pairtools* facilitates statistical analysis and visualization. *Pairtools* represents a versatile new foundation for studying principles of 3D genome organization and their impacts on gene regulation and cellular phenotypes.

## Intro

Chromosome conformation capture technologies (3C+), particularly Hi-C, revolutionized the study of genome folding by using high-throughput sequencing to measure spatial proximity. All 3C+ protocols involve five steps: (i) chemical cross-linking of chromatin [1], (ii) partial digestion of DNA, (iii) DNA ligation, (iv) library preparation (i.e. ultrasonication, purification, and amplification), and (iv) sequencing [2]. Ligation is the pivotal step that records the spatial proximity of DNA loci as libraries of chimeric DNA molecules. The resulting libraries are typically sequenced in short-read paired-end mode (around 50–300 bp on each side), producing millions to tens of billions of sequencing reads.

3C+ data are typically computationally processed in three stages (Fig 1a), each requiring specialized computational tools. First, sequencing reads are *aligned* to the reference genome. Next, pairs of genomic locations are *extracted* from the alignments. These pairs may be interpreted as genomic contact events. For various statistical and technical reasons, pairs are normally aggregated or *binned* to form contact matrices at various lower genomic resolutions (for bin-free methods of aggregation, see [3,4]). Binned contact maps can be stored, manipulated, and analyzed using downstream tools and software packages, such as *cooler* [5] and *cooltools* [6] from Open2C. While 3C+ data is often analyzed as binned contact matrices, important quality control information, such as exact mapping positions, strand orientation, mapping quality, and pair type, require analysis at the level of pairs. For these reasons, it is important to be able to flexibly generate, interpret, store, and manipulate pairs-level data.

The rapid adoption of 3C+ technologies by the genomics community poses two major computational challenges. First, the quantity of 3C+ data is increasing rapidly. A growing number of labs and consortia (4DN [7], ENCODE [8], DANIO-CODE DCC [9]) use Hi-C to produce large quantities of proximity ligation data. At the same time, new protocols, such as Micro-C [10] and Hi-C 3.0 [11], improve resolution and sensitivity and generate even larger datasets. This requires software to be fast, parallelizable, and storage-efficient. Second, emerging 3C+ variants introduce a growing diversity of protocols. This includes methods to measure contacts within individual homologs [12], sister chromatids [13,14], single cells [15–19], and multi-way contacts (MC-4C [20], Tri-C [21], MC-3C [22], Pore-C [23], and Nano-C [24]). The growing variety of 3C+ methods thus requires software to be versatile and flexible.

Here we introduce *Pairtools*, a suite of flexible and performant tools for converting sequenced 3C+ libraries into captured chromosome contacts. *Pairtools* provides modules to (i) extract and classify pairs from sequences of chimeric DNA molecules, (ii) deduplicate, filter, and manipulate resulting contact lists, and (iii) generate summary statistics for quality control (QC). *Pairtools* enables the processing of standard Hi-C [2] as well as many Hi-C-derived protocols, including homolog- and sister-sensitive, and single-cell Hi-C protocols. *Pairtools* relies on standard data formats and serves as a crucial link between sequence aligners and Hi-C contact aggregation tools, together forming highly efficient pipelines for Hi-C data processing (Fig 1a and 1b). (Fig 1a and 1b). Benchmarks against several popular 3C+ data mappers show advantages of *pairtools* for high-performance and flexible 3C+ analysis. *Pairtools* is implemented in Python, powered by common data analysis libraries such as NumPy [25] and

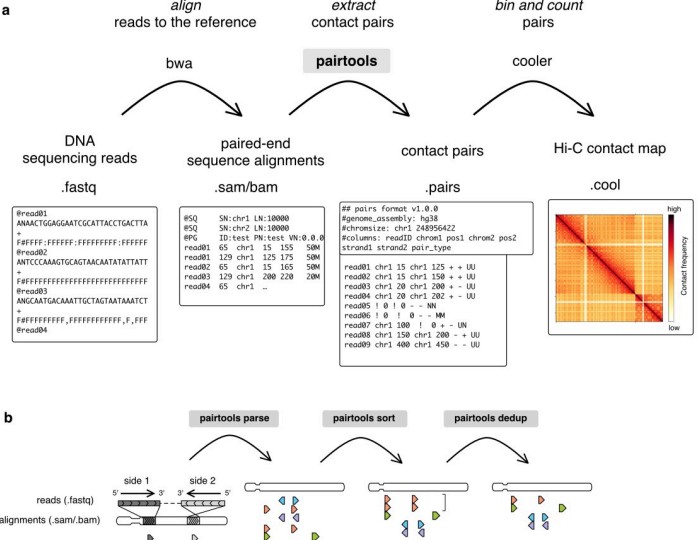

**Fig 1. Processing 3C+ data using pairtools. a**. Outline of 3C+ data processing leveraging *pairtools*. First, a sequenced DNA library is mapped to the reference genome with sequence alignment software, typically using *bwa mem* for local alignment. Next, pairtools extracts contacts from the alignments in.sam/.bam format. *Pairtools* outputs a tab-separated.pairs file that records each contact with additional information about alignments. A.pairs file can be saved as a binned contact matrix of counts with other software, such as *cooler*. The top row describes the steps of the procedure; the middle row describes the software and chain of files; the bottom row depicts an example of each file type. **b**. Three main steps of contact extraction by *pairtools*: *parse*, *sort*, and *dedup*. *Parse* takes alignments of reads as input and extracts the pairs of contacts. In the illustration, alignments are represented as triangles pointing in the direction of read mapping to the reference genome; each row is a pair extracted from one read. The color represents the genomic position of the alignment with the smallest coordinate, from the leftmost coordinate on the chromosome (orange) to the rightmost coordinate on the chromosome (violet). *Sort* orders mapped pairs by their position in the reference genome. Before sorting, pairs are ordered by the reads from which they were extracted. After sorting, pairs are ordered by chromosome and genomic coordinate. *Dedup* removes duplicates (pairs with the same or very close positions of mapping). The bracket represents two orange pairs with very close positions of mapping that are deduplicated by *dedup*.

pandas [26], offers a CLI, and is available as open-source software at: https://github.com/open2c/pairtools/.

## Design and implementation

*Pairtools* provides tools for each step of data processing between the sequence alignment and contact binning (Fig 1a and 1b): extraction, sorting, deduplication, filtering, and quality control of contact pairs.

   *Pairtools* adheres to the following design principles, aligned with Unix style principles [27]:

- Split functionality into tools that can be used independently or combined into pipelines.

- Focus on modularity, flexibility, and clarity first and performance second. Data processing should be as fast as alignment, but not necessarily faster.

- Outsource functionality when possible. Rely on existing software for alignment and work-flow managers for data pipelining.

- Leverage the rich ecosystem of Python and data analysis libraries, including *NumPy* [25], *pandas* [28], *scipy* [29] and *scikit-learn* [30].

- Use a standardized tabular format for pairs.

- Accommodate existing Hi-C protocol modifications by generalizing existing tools. When not possible, introduce protocol-specific tools.

- Take advantage of multi-processing and streaming to improve performance.

## Results

### Essential building blocks for 3C+ pair processing

*Pairtools* processes 3C+ data in three essential steps (Fig 1b). First, the genomic alignments of 3C+ products are *parsed* into individual contact events, or *pairs*. Second, the resulting pairs are *sorted* to facilitate data access and analysis. Third, pairs are *deduplicated*, resulting in the final list of 3C+ contacts.

The minimal *pairtools*-based pipeline is expressed concisely as:

```
bwa mem -SP index input.R1.fastq input.R2.fastq | \

pairtools parse -c chromsizes.txt |

pairtools sort | \

pairtools dedup | \

cooler cload pairs -c1 2 -p1 3 -c2 4 -p2 5 chromsizes.txt:1000—
  output.1000.cool
```

Below, we describe these three steps and the corresponding *pairtools* functionality in detail.

**Parse: Extracting single proximity ligation events.**  The DNA molecules in a 3C+ library are chimeric by design: spatial proximity between different genomic segments is captured as DNA ligation events, which are then read out via DNA sequencing. *Pairtools* makes use of existing software for alignment, taking.sam/.bam files [31] as input. Sequence alignments in. sam/.bam comprehensively describe the structure of chimeric DNA molecules in the 3C + library. Each entry in these files stores an alignment of a continuous read segment to the reference genome. Entries include mapping position, as well as flags and tags describing mapping properties (such as uniqueness of mapping, nucleotide variations, error probability, and more). The properties can be read with tools like *pysam* [32]. However, alignments in the.sam/ .bam files are reported sequentially and are not structured as contact pairs. Extracting proximity ligation events from alignments requires additional processing, which *pairtools* achieves with *parse*.

*Pairtools parse* is developed and optimized for Hi-C libraries with chimeric DNA molecules formed via single ligation events. *Pairtools* is designed to analyse the output of local sequence aligners (e.g., bwa mem). Local aligners can align subsequences of the input sequence to multiple locations in the genome, whereas global aligners expect the whole input sequence to align to one location of the genome. This feature of local aligners is better suited to the chimeric molecules generated by 3C+ protocols. **Pairtools parse** extracts pairs of alignments that are adjacent in the chimeric molecules and reports them as contacts between two loci in the genome (S1a–S1c Fig). *Parse* also detects cases when one of the DNA fragments in a pair is sequenced on both sides of the read, producing two distinct alignments. As these two alignments do not represent a contact *parse* merges them and "rescues" the true contact pair (S1d Fig). The output of parse adheres to the standard format.pairs [33] (discussed below).

The engine of **pairtools parse** uses *pysam* [32] to extract tags and flags from.sam/.bam files. **Pairtools parse** can run in combination with a variety of popular local sequence aligners, such

as *bwa mem* [34], *bwa mem2* [35], *minimap2* [36], and others, as long as their output complies with the.sam/.bam format. Importantly, for 3C+ data aligners must align the two reads of a pair independently (i.e., avoid 'pair rescue'). In the case of bwa mem, adding the -SP flags ensures this behavior.

**Pairtools uses and extends the.pairs format.** *parse* output contact tables in a text tab-separated format called.pairs, designed by the NIH 4DN consortium [33]. As a text format,. pairs has several advantages over custom binary formats: (i) text tables can be processed in all programming languages, (ii) are easily piped between individual CLI tools, and (iii) have a set of highly efficient utilities for sorting (Unix *sort*), compression (*bgzip/lz4*), and random access (*tabix* [37] /*pairix* [33]).

Each tab-separated row in a.pairs file describes a single observed contact. The required columns contain the id of the read and the genomic locations of the two sites that formed the contact. **Pairtools parse** augments these data with the pair type (S1a–S1g Fig) and optional columns with details of the genomic alignments supporting the contact.

Headers of.pairs files can store metadata, which by default includes the names of columns and the description of chromosomes of the reference genome. To ensure data provenance, *pairtools* extends this standard header with (i) the header of the.sam files that stored the original alignments and (ii) the complete history of data processing, with a separate entry for each CLI command of *pairtools* that was applied to the file. *Pairtools* provides a set of CLI and Python functions to parse and manipulate the header. **Pairtools header** can generate a new header, validate an existing one, transfer it between.pairs files, as well as set new column names (Fig 2a). **Pairtools header** helps to fix.pairs files that were generated by scripts or software that do not fully comply with the.pairs format specifications and have missing or improperly formatted headers.

**Sort and flip: Organizing contact lists.** Sorting the pairs in contact tables facilitates data processing, and analyses as it (i) enables fast random access via indexing and (ii) simplifies the detection of duplicated pairs, as they end up in adjacent rows of the table. *Pairtools* sorts pairs in two steps. First, individual pairs are "flipped," i.e., arranged so that the alignment with the lower coordinate in the genome comes first in the pair (Fig 2b). Flipping ensures a reproducible sorting for data indexing and duplicate removal. Flipping is performed by default during parsing and can be done manually by **pairtools flip**. Second, **pairtools sort** performs Unix-based sorting of pairs in contact tables according to their genomic positions (on chrom1, chrom2, pos1, pos2). This sorting scheme has multiple advantages: it arranges pairs in blocks corresponding to contacts between a given pair of chromosomes, separates within (*cis*) from between (*trans*) chromosome contacts, and facilitates access to unmapped and multi-mapped pairs.

**Dedup: Detecting duplicated DNA molecules.** A key issue for sequencing-based protocols, including Hi-C and other 3C+, is that the same DNA product can be duplicated by PCR and then sequenced and reported more than once, thus introducing an error into their quantitative measurements. *Pairtools* provides a computationally efficient tool for duplicate removal called **pairtools dedup** (Fig 1b). It detects clusters of pairs with matching genomic coordinates and strands and removes all but one pair. To accommodate non-standard protocols, e.g. some varieties of single-cell Hi-C [17,18], where the amplification step occurs between ligation and sonication, **pairtools dedup** can detect duplicated pairs even when their mapped positions do not match exactly. To enable such imperfect coordinate matching, **pairtools dedup** relies on a KD-tree-based fixed radius nearest neighbor search algorithm [38], implemented in *scipy* and *scikit-learn*. To reduce its memory footprint, **pairtools dedup** processes input data in overlapping chunks. Finally, **pairtools dedup** can take additional columns from a.pair file and require additional properties of pairs to match, such as type of contact (direct/indirect), presence of mutations, or phased haplotype [12]).

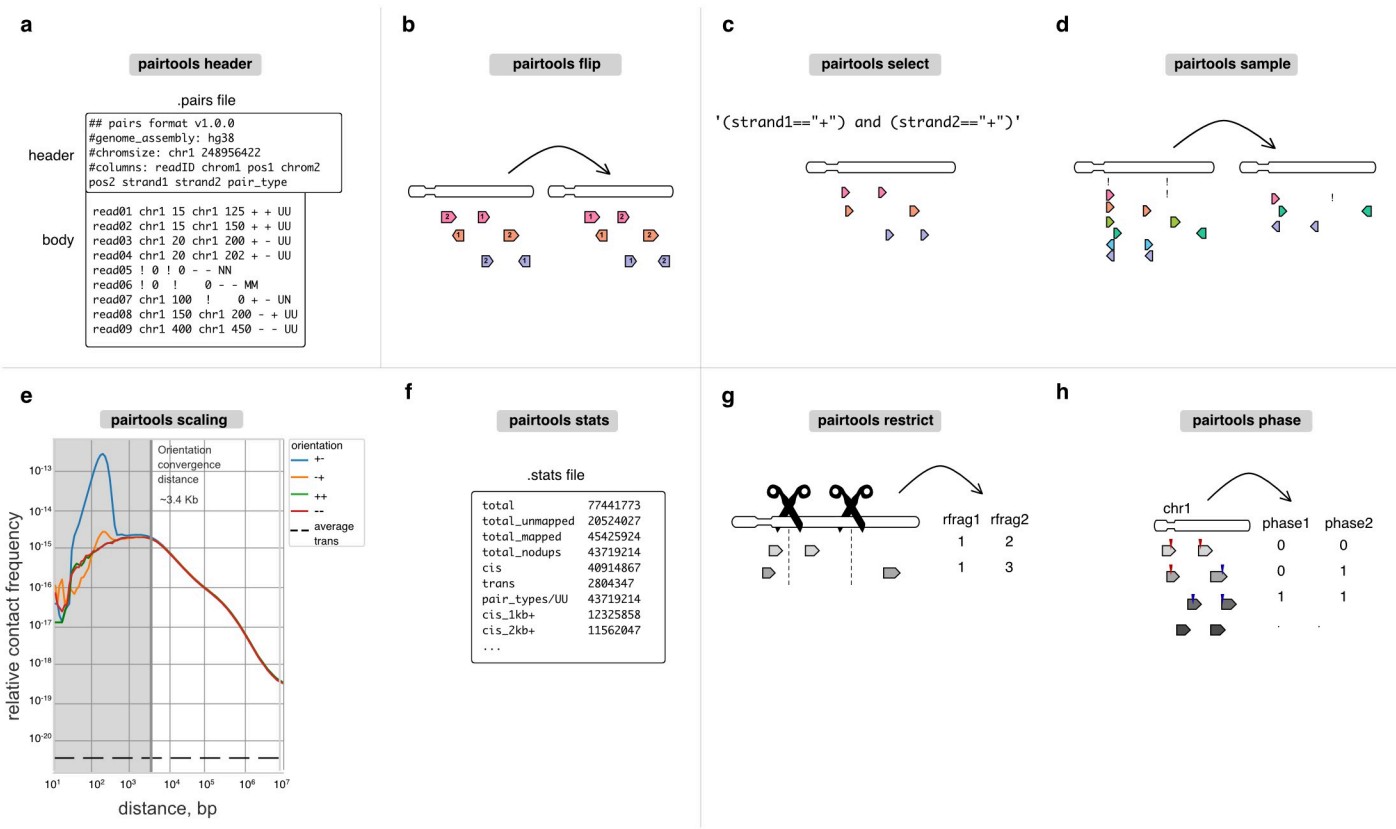

**Fig 2. Auxiliary tools for building feature-rich pipelines. a**. *Header* verifies and modifies the.pairs format. **b-d**. *Flip*, *select*, and *sample* are for pairs manipulation. **e-f**. *Scaling* and *stats* are used for quality control. For scaling, we report scalings for all pairs orientations (+-, -+, ++, —) as well as average trans contact frequency. ***Orientation convergence distance*** is calculated as the last rightmost genomic separation that does not have similar values for scalings at different orientations. **g-h**. *Restrict* and *phase* are protocol-specific tools that extend *pairtools* usage for multiple 3C+ variants.

Tracking duplicates with *pairtools* enables an estimate of *library complexity*, i.e. the total number of unique DNA molecules prior to PCR amplification, an important QC for 3C+. Library complexity can guide the choice of sequencing depth of the library and provide an estimate of library quality. To estimate library complexity, *pairtools* assumes that each sequencing read is randomly chosen with replacement from a finite pool of fragments in DNA library [39,40].

## Tools for building feature-rich 3C+ pipelines

In addition to supporting the parse-sort-dedup steps (Fig 1b) that are sufficient to build a minimalistic 3C+ processing pipeline, *pairtools* also provides tools to build feature-rich pipelines. This includes tools for automated QC reporting, filtering of high-quality 3C+ data, as well as merging of replicates and conditions into meta contact maps, required for a complete and convenient end-to-end data processing pipeline.

**Select, sample, and merge: Manipulating pairs.**    *Pairtools* provides a collection of tools for the manipulation of tabular pairs data.

- ***pairtools select*** splits and subsets datasets according to arbitrary filter expressions (Fig 2c). These expressions are provided as Python functions, enabling expressive and powerful

filters. Filters can include wildcard and regex matching on string columns, custom functions from 3rd-party libraries (for examples of advanced usage, see Sister-C [14], scsHi-C [13]), as well as filtering pairs to a given subset of chromosomes.

- ***pairtools sample*** can generate random subsets of pairs, e.g., to equalize coverage between libraries or assess the statistical robustness of analyses (Fig 2d).

- ***pairtools merge*** combines multiple input datasets into one; for pre-sorted inputs, it efficiently produces sorted outputs.

Together, ***pairtools select*** and ***merge*** enable the split-apply-combine pattern for distributed data processing.

**Stats and scaling: Quality control.**   3C+ experiments have multiple failure modes and thus require tight quality controls. Many experimental issues can be inferred from the statistics of the resulting 3C+ data (for a detailed discussion, see [41,42]).

A particularly rich source of information about 3C+ experiments is the decay of contact frequency with the genomic distance referred to as the **P(s)** [2] curve or **scaling** (borrowing the physics terminology for power-law relationships). Scalings are used both to characterize mechanisms of genome folding [6] and reveal issues with QC [1]. For example, early flattening of the scaling in Micro-C revealed the importance of long cross-linkers [10]. Scalings can also be used to determine that combinatorial expansion of walks produces undesirable contacts because indirect contacts result in flatter scaling (S2e Fig) [22,23].

***Pairtools scaling*** calculates **strand-oriented scalings** that can be used for by-product quality control and filtration (Fig 2e). After the ligation step, some fragments can form a valid pair or produce unwanted 3C+ by-products, such as self-circles, dangling ends (unligated DNA) (S2c Fig), and mirror reads (potential PCR artifacts) [43]. A short-range peak in divergent orientation is a sign of self-circled DNA, while a short-range peak in convergent orientation is a sign of dangling ends (Figs 2e and S2d) [41,42]. For example, early Hi-C variants with a low concentration of cross-linker caused the prevalence of self-circles [44]. At larger genomic separations, pairs are formed in all four orientations with equal probabilities, and strand-oriented scalings converge. The ***orientation convergence distance*** indicates the minimum distance where pairs can simply be interpreted as contacts for a 3C+ dataset. Removing contacts below the orientation convergence distance removes nearly all by-products marked by restriction fragment annotation (see below, S2b Fig). For DpnII Hi-C datasets, orientation convergence usually occurs by ~2kb. We note that for analyses downstream of QC, scaling can also be calculated from corrected binned data, e.g., using cooltools [6].

For convenience and workflow reproducibility, ***pairtools stats*** automatically reports genome-wide contact scalings. It also generates additional summary statistics, including the total number of pairs of each type, the number of *trans* contacts between different chromosomes, as well as the orientation convergence distance in cis. This information has been used to understand the impact of various protocol decisions. For example, information about the frequency of *trans* and different ranges of *cis*-contacts demonstrated that extra cross-linking yields more intra-chromosomal contacts [1]. The frequency of contacts between the nuclear and mitochondrial genomes reflects the noise introduced by various digestion strategies [1].

***pairtools stats*** produces a human-readable nested dictionary of statistics stored in a YAML file or a tab-separated text table (used in [45,46]) (Fig 2f). These outputs can be visualized with MultiQC [47], an interactive web-based tool that aggregates a wide set of sequencing QC statistics and provides an overview of whole collections of samples. The orientation convergence distance reported by ***pairtools stats*** can also be used to remove all Hi-C byproducts from binned output conservatively:

```
pairtools stats library.nodups.pairs.gz -o library.stats

CONV_DIST=`grep "summary/dist_freq_convergence/convergence_
  dist" library.stats | cut -f2`

pairtools select "(chrom1!=chrom2) or (abs(pos1-pos2)>=${CONV_
  DIST})" library.nodups.pairs.gz \

| cooler cload pairs -c1 2 -p1 3 -c2 4 -p2 5 chromsizes.txt:1000
  —output.1000.cool
```

Such filtering is, however, typically unnecessary as cooler and cooltools by default ignore the first two diagonals in all computations. This filter is sufficient to remove by-products of 4bp-cutter Hi-C and Micro-C at resolutions > = 1kb.

## Protocol-specific tools

Chromosome capture is a growing field, with novel protocol modifications emerging regularly [48]. Thanks to its flexible architecture, *pairtools* can process data from many such experiments. For example, data from chemical modification-based protocols, such as scsHi-C [13], sn-m3C-seq [49], or Methyl-HiC [50] can be processed by (i) extracting sequence mismatches into separate columns of.pairs by ***pairtools parse*** and (ii) filtering and subsetting pairs based on these columns with ***pairtools select*** (Fig 2c). For other popular and actively developing protocol variants, such as Micro-C [10], haplotype-resolved [12] and single-cell Hi-C [51], *pairtools* provides specialized utilities.

**Restrict: Annotating pairs by restriction fragments.** Many 3C+ protocols, particularly original Hi-C, rely on cutting DNA by restriction enzymes and theoretically should generate ligations only between restriction sites [41,52]. Thus, early 3C+ analysis pipelines included filters that detected and removed (i) unligated single restriction fragments and (ii) ligations between pieces of DNA located far from any restriction sites. ***Pairtools restrict*** enables such filters by assigning the nearest restriction sites to each alignment in a pair (Fig 2g).

However, we find restriction-based filters unnecessary for more recently published Hi-C and do not include them in the standard *pairtools* pipeline. First, in our tests of recently published Hi-C datasets [1], the statistical properties of pairs located far from and close to restriction sites proved nearly the same (S2a Fig). Second, we found that unligated pieces of DNA can be removed by a simpler filter against short-distance pairs, which can be calibrated using strand-oriented scalings [42] (S2b Fig). For downstream analyses in cooler [5], such by-products are removed by dropping pairs of bins with separations below a cutoff distance, which corresponds to removing a few central diagonals of a binned contact matrix. Finally, the annotation of restriction sites becomes less accurate and unproductive for libraries generated with frequent and/or flexible cutters (e.g., DpnII, MboI, and DdeI), cocktails thereof, and impossible in restriction-free protocols, such as Micro-C [10] and DNase Hi-C [53].

**Phase: Annotating haplotypes.** Haplotype-resolved Hi-C [12,19,54–56] leverages sequence variation between homologous chromosomes to resolve their contacts in *cis* and *trans*. In particular, single nucleotide variants (SNVs) can be used to differentiate contacts on the same chromosome (*cis*-homologous) from contacts between two homologs (*trans*-homologous).

***Pairtools phase*** is designed to resolve reads mapped to maternal and/or paternal genomes (Fig 2h). To enable analyses with ***pairtools phase***, reads must be mapped to a reference genome that contains both haplotypes, reporting suboptimal alignments; these suboptimal alignments will be extracted into separate.pairs columns by ***pairtools parse***. By checking the

scores of the two best suboptimal alignments, **pairtools phase** distinguishes true multi-mappers from unresolved pairs (i.e., cases without a distinguishing SNV/indel) and reports the phasing status of each alignment in a pair: non-resolved, resolved as the first haplotype or the second haplotype, or multi-mapper. Further downstream, **pairtools select** and **pairtools stats** enable splitting and analyzing phased pairs.

The approach of pairtools to resolving homologs is designed to minimize homolog biases: when reads are aligned to both homologs simultaneously, both homologs are treated equally. For example, in the two studies that introduced the approach behind pairtools phase [12,54], the observed homolog bias was around 1–3%. Similarly important for this low bias was the fact that this study re-sequenced both parental strains as well as the genome of their F1 progeny, thus obtaining high-quality SNVs for both homologs. Equally important for minimizing the bias, this study sequenced both parental strains and their F1 progeny, yielding high-quality single nucleotide variants (SNVs) for both homologs. Otherwise, potential differences in the SNV quality between the two homologs (for example, if two parental strains were sequenced with different quality) could introduce homolog bias.

**Filterbycov: Cleaning up single-cell data.**   Single-cell 3C+ experimental approaches shed light on variation and regularities in chromatin patterns among individual cells [51]. Single-cell 3C+ data can be processed with *pairtools* almost the same way as bulk Hi-C, with the addition of one filter. In single cells, the number of contacts produced by each genomic site is limited by the chromosome copy number. Thus, regions with irregularly high coverage indicate amplification artifacts or copy number variations [18,51] and must be excluded from downstream analyses [17,57]. **Pairtools filterbycov** calculates genome-wide coverage per cell and excludes regions with coverage exceeding the specified threshold. This procedure helped to remove regions with anomalous coverage in single-cell Hi-C studies in *Drosophila* [18].

## Performance and comparison with other tools

Contact extraction from raw sequencing data is the first and typically most time-consuming step of the 3C+ data processing. *Pairtools* is one of the fastest methods (lagging behind only *Chromap* [58]) without consuming more memory (in combination with *bwa mem*), making it the best candidate for scalable 3C+ data processing (Fig 3). Notably, *Pairtools* is the only tool that combines high performance with the flexibility to enable adaptations to a broad range of 3C+protocols (Table 1).

## Discussion

*Pairtools* provides a set of interoperable and high-performance utilities for processing 3C+ contact data [6,67,68], particularly for extraction of contacts from sequence alignments, manipulating pair files, deduplication, data filtering, generation of summary statistics and contact scalings, quality control, and treatment of data generated with 3C+ protocol modifications.

*pairtools* is easy to install via *pip* and *conda*. We provide extensive documentation of *pairtools* [69], including example scripts of minimal *pairtools*-based pipelines and Jupyter tutorials with explanations of the working of *pairtools*, restriction, and phasing in *pairtools* GitHub repository [70].

The modular structure of *pairtools* and its usage of the.pairs format [33] already make it useful in many pipelines. *pairtools* is used in the 4DN pipeline (standard Hi-C) [7], the *PORE-C* pipeline (multi-way Hi-C) [71], HI-CAR *nf-core* pipeline (open-chromatin-associated contacts) [72], and iMARGI pipelines (RNA-DNA contacts) [45,73]. *pairtools* also serve as the foundation of *distiller* [74], a feature-rich fastq-to-cooler [5] pipeline, based on the nextflow workflow framework [75] and maintained by Open2C [68]. *Distiller* takes advantage of piping

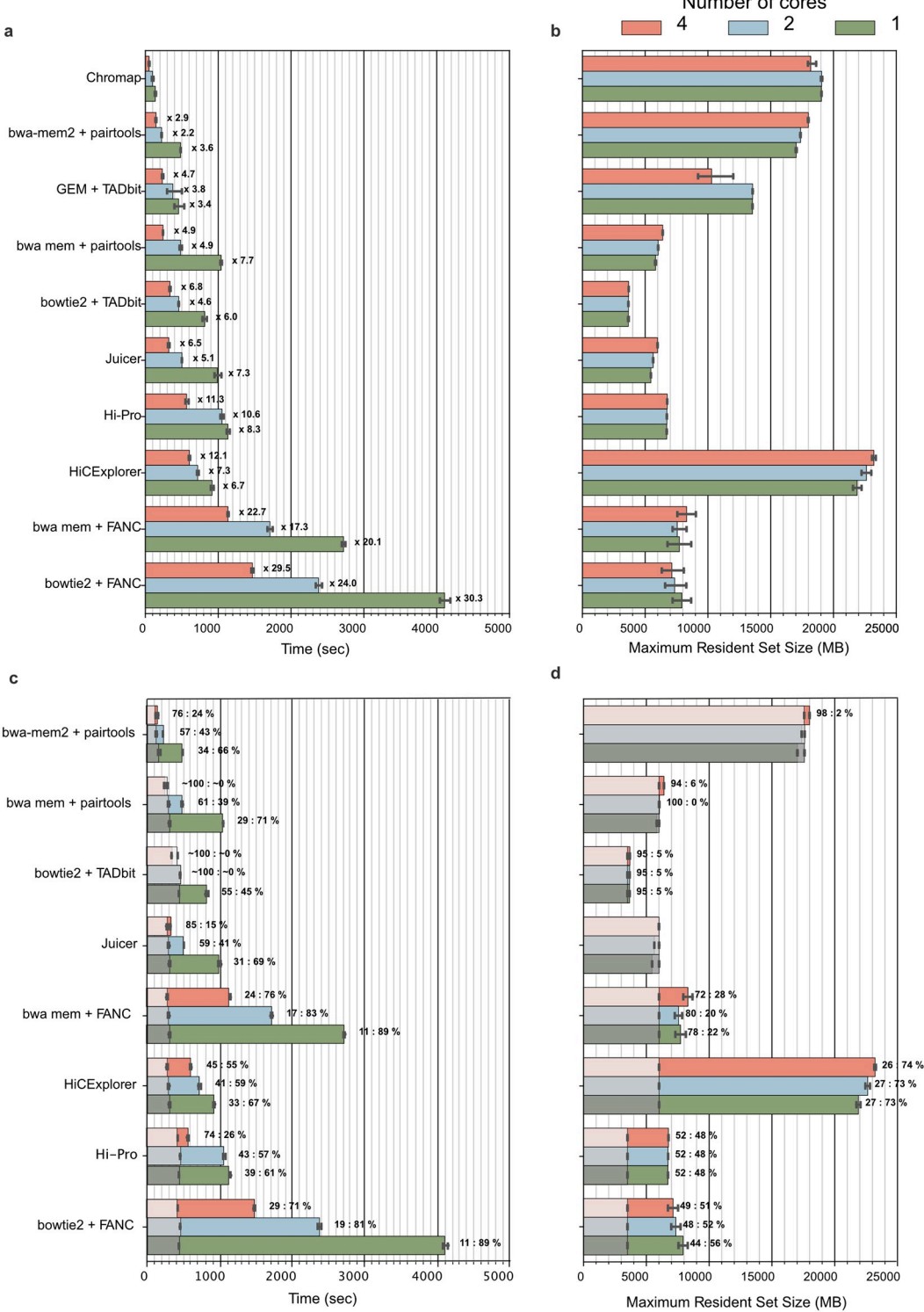

**Fig 3. Benchmark of different Hi-C mapping tools for one mln reads in 5 iterations (data from [64]). a**. Runtime per tool and number of cores. The labels at each bar of the time plot indicate the slowdown relative to *Chromap* [58] with the same number of cores. **b**. Maximum resident set size for each tool and number of cores. **c**. Runtime per tool and number of cores compared to the runtime of the corresponding mapper (gray shaded areas). Labels at the bars reflect the percentage of time used by the mapper versus the time used by the pair parsing tool. **d**. Maximum resident set size for each tool and number of cores compared with that of the corresponding mapper. To make the comparison possible, the analysis for each tool starts

with.fastq files, and the time includes both read alignment and pairs parsing. For *pairtools*, we tested the performance with regular *bwa mem* [34] and *bwa mem2* [35], which is ~2x faster but consumes more memory. Note that for *HiC-Pro*, we benchmark the original version and not the recently-rewritten *nextflow* [65] version that is part of *nf-core* [66]. *FANC*, in contrast to other modular 3C+ pairs processing tools, requires an additional step to sort.bam files before parsing pairs that we include in the benchmark. For *Juicer*, we use the "early" mode. *Chromap* is not included in this comparison because it is an integrated mapper [58].

*pairtools* command outputs and can parallelize 3C+ data processing within a single machine, on a cluster, or in the cloud.

In the future, a binary storage format for pairs could substantially speed up 3C+ contact extraction. Currently, *zarr* [76] is the best candidate as it allows variable length strings (not supported by *hdf5* [77]) and allows appending columns and storing multiple tables in a single file (not supported by *parquet* [78]).

To summarize, *Pairtools* provides an adaptable framework for future development to enable the expanding universe of 3C+ protocols.

## Availability and future directions

Open-source code is freely available at https://github.com/open2c/pairtools. Additional documentation is available at https://pairtools.readthedocs.io/, with interactive tutorials.

Pairtools is integrated into the high-performant nextflow-based pipeline *distiller*: https://github.com/open2c/distiller-nf/.

Code for generating manuscript figures available at: https://github.com/open2c/open2c_vignettes/tree/main/pairtools_manuscript/. Benchmark is available at https://github.com/open2c/pairtools/tree/master/doc/examples/benchmark/.

**Table 1. Qualitative comparison of the tools for pairs extraction from 3C+ sequencing data.** We consider methods modular if they have multiple tools that can be used separately or combined in a custom order. HiCExplorer is modular, but its tool for contact extraction is not (indicated with *). We consider methods flexible if they allow parameterization of data processing (e.g., restriction enzyme). We do not consider control only over technical parameters, like the number of cores, to be flexible. For restriction sites, we consider whether a method can either annotate or filter by restriction site.

| Tool | Short description | Input | Output | Modular | Flexible | Aligner | Restriction sites | Quality control | Support for modified 3C + protocols |
|---|---|---|---|---|---|---|---|---|---|
| Pairtools | python API/ CLI tools | .sam alignments | pairs | Yes | Yes | bwa mem, bwa mem2, minimap2, bwa aln | Yes | Aggregated stats, scaling | Haplotype-resolved; chemical modification-based; multi-contact; single-cell |
| Chromap [58] | single executable | .fastq reads | contact maps | No | No | chromap | No | No | No |
| Juicer [59] | java/shell script pipeline | .fastq reads | pairs and contact maps | No | Yes | bwa mem | Yes | Aggregated stats | Haplotype-resolved |
| HiC-Pro [60] | python/R/shell script pipeline | .fastq reads | pairs and contact maps | No | Yes | bowtie2 | Yes | QC report | Haplotype-resolved |
| HiCExplorer [61] | python API/ CLI tools | .sam alignments | contact maps | Yes * | Yes | bwa mem | Yes | QC report | No |
| FANC [62] | python API/ CLI tools | .fastq reads or. sam alignments | pairs | Yes | Yes | bwa mem, bowtie2 | Yes | QC report | No |
| TADbit [63] | python API/ CLI tools | .fastq reads or. sam alignments | parsed reads or contact maps | Yes | Yes | gem, bowtie2, hisat | Yes | Yes | No |

We welcome issues and questions on GitHub https://github.com/open2c/pairtools/. For questions about the following parts of the repository, please tag the relevant contributors on GitHub.

Pairtools parse: AG @golobor, AAG @agalitsyna

Pairtools dedup: IMF @phlya, AAG @agalitsyna

Pairtools stats, scaling, filtering by coverage, restriction, phasing and other protocol-specific tools: AG @golobor, AAG @agalitsyna

## Supporting information

**S1 Fig. Walks policies for pairs parsing. a-h**. Different types of paired-end reads processed by *parse*. Notation is the same as in Fig 1b. **a**. Single contact with two alignments. Each side of the read contains a uniquely mapped alignment (red and blue). **b**. Unmapped pairs. Either one (top) or both (bottom) sides of the read do not contain segments aligned to the reference genome. **c**. Multiple mapped pairs. Either one (top, center) or both (bottom) sides of the read contain a segment that is mapped to multiple locations in the genome. **d**. Single contact with three alignments. One side of the read pair contains two segments that align to different genomic locations (red on the 5' end and green on the 3'), while the second read side contains only one alignment (blue). If the green and blue alignments have opposite orientation, are located on the same chromosome and separated by the distance shorter than the typical molecule size, pairtools parse considers them part of the same DNA fragment. *parse* recognize "rescues" such pairs reports them as a contact between the red and the blue alignments. **e**. An "unrescuable" molecule with three alignments highlights the difference between walks policies. One side of the reads contains two unique alignments to distinct genomic locations (red and green). If the 3' alignment (green) and the 2nd side alignment (blue) are too distant, do not have convergent orientations, or are in trans, the molecule cannot be "rescued" into a single contact and instead is considered as a two-contact walk.—*walks-policy mask* ignores such cases (W).—*walks-policy all* reports both ligations. '5unique'and '5any'report the two 5'-most alignments at each read side. '3unique'and '3any'report the 3'-most alignments at each read side. **f**. A molecule formed via three ligations. Both sides of the read contain two segments mapped to different unique locations.—*walks-policy mask* ignores such cases (W).—*walks-policy all* reports all three ligation events.—*walks-policy 5unique* and *5any* report the two 5'-most alignments at each read side.—*walks-policy 3unique* and *3any* report the two 3'-most alignments at each read side. **g**. Readthrough, i.e. the case when the sum of read length exceeds the molecule length, produces internally duplicated alignments. In this example, the molecule is formed via three ligations between four DNA fragments. However, the readthrough produces duplicated alignments on the 3' ends of both sides, resulting in contains six alignments in total.—*walks-policy all* recongizes such events and correctly reports three ligation events. **h**. A scenario is presented where a molecule is potentially formed through two ligations between three DNA fragments, with the 5'-most fragment remaining unmapped. Due to this unmapped fragment,—*walks-policy 5any* reports a null-unique pair, in contrast to—*walks-policy 5unique*, which selects the 5'-most unique alignment on the left side (represented in green).
(TIFF)

**S2 Fig. Pairtools scaling and quality control of 3C+ data. a**. Orientation-dependent scalings for pairs grouped by distance to the nearest restriction site (DpnII Hi-C from [1]). Scalings are very close at genomic separations beyond the orientation convergence distance. **b**. Generation of normal pairs and by-products in 3C+ protocol. Normal pairs originate from distinct restriction fragments separated by at least one restriction site (in black). Pairs in self-circles and dangling ends are located on the same restriction site, either in divergent (self-circles) or

convergent (dangling ends) orientation. **c**. Counts of pairs are categorized into four groups: regular pairs, dangling ends, self circles, and mirror pairs [43] for a test sample of 11 million pairs, by restriction enzyme annotation (columns) and convergence distance (rows). For restriction enzyme annotation, we considered dangling ends to be mapped to the same restriction fragment in the convergent orientation, self circles in the divergent orientation, and mirror pairs in the same orientation. For convergence distance annotation, we conservatively considered all the pairs below convergence distance as potential by-products and assigned them to each category by their orientation as for the restriction enzyme annotation. Both methods produce highly congruent filtration, as seen by the relatively smaller number of off-diagonal pairs. **d**. Scaling with prominent peak of self-circles and dangling ends. A short-range peak in pairs mapped to opposing strands facing away from each other (divergent) is a sign of self-circled DNA, while a short-range peak in pairs mapped to opposing strands facing each other (convergent) pairs is a sign of dangling ends. **e**. Scalings for direct, indirect (2- and 3-hops), and unobserved contacts. Note that multi-hop contacts have a flatter scaling, potentially indicating more ligations in the solution [22,23].
(TIFF)

## Acknowledgments

The authors thank Leonid Mirny, Job Dekker and members of the Center for 3D Structure and Physics of the Genome for feedback on tool functionality. All authors made contributions as detailed in the Open2C authorship policy guide. All authors are listed alphabetically, read, and approved the manuscript.

## Author Contributions

**Conceptualization:** Aleksandra A. Galitsyna, Anton Goloborodko.

**Formal analysis:** Nezar Abdennur, Geoffrey Fudenberg, Ilya M. Flyamer, Aleksandra A. Galitsyna, Anton Goloborodko, Maxim Imakaev, Sergey V. Venev.

**Investigation:** Nezar Abdennur, Geoffrey Fudenberg, Ilya M. Flyamer, Aleksandra A. Galitsyna, Anton Goloborodko, Maxim Imakaev, Sergey V. Venev.

**Methodology:** Nezar Abdennur, Geoffrey Fudenberg, Ilya M. Flyamer, Aleksandra A. Galitsyna, Anton Goloborodko, Maxim Imakaev, Sergey V. Venev.

**Software:** Nezar Abdennur, Geoffrey Fudenberg, Ilya M. Flyamer, Aleksandra A. Galitsyna, Anton Goloborodko, Maxim Imakaev, Sergey V. Venev.

**Visualization:** Nezar Abdennur, Geoffrey Fudenberg, Ilya M. Flyamer, Aleksandra A. Galitsyna, Anton Goloborodko, Maxim Imakaev, Sergey V. Venev.

**Writing – original draft:** Aleksandra A. Galitsyna, Anton Goloborodko.

**Writing – review & editing:** Nezar Abdennur, Geoffrey Fudenberg, Ilya M. Flyamer, Aleksandra A. Galitsyna, Anton Goloborodko, Maxim Imakaev, Sergey V. Venev.

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
