## [Decision Letter · Decision Letter 0]

28 Nov 2023

Dear Dr. Goloborodko,

Thank you very much for submitting your manuscript "Pairtools: from sequencing data to chromosome contacts" for consideration at PLOS Computational Biology.

As with all papers reviewed by the journal, your manuscript was reviewed by members of the editorial board and by several independent reviewers. In light of the reviews (below this email), we would like to invite the resubmission of a significantly-revised version that takes into account the reviewers' comments.

We cannot make any decision about publication until we have seen the revised manuscript and your response to the reviewers' comments. Your revised manuscript is also likely to be sent to reviewers for further evaluation.

Sincerely,

Ferhat Ay, Ph.D

Academic Editor

PLOS Computational Biology

Jian Ma

Section Editor

PLOS Computational Biology

Reviewer's Responses to Questions

**Comments to the Authors:**

**Reviewer #1:** The Open2C consortium presents a manuscript in which they describe a computational tool (pairtools) that parses sequencing data from a Hi-C experiment that has been mapped using a standard pipeline (e.g. bwa or minimap). It parses the raw mapping file that can be used as input for cooler, which can generate Hi-C contact matrices that are used for further analysis and viewing of the Hi-C data. In that sense it serves a function similar to SAMtools. The paper is clear summary of what the software can do. There is also a nice comparison to similar tools, including memory and speed benchmarking.

I have no major comments for the paper. I have a few suggestions for improvement.

- Can the authors discuss whether pairtools phase has a bias for the reference allele. This can be an issue with allele-specific reads and specific tools have been developed to counteract this (e.g. WASP, PMID: 26366987). I can imagine that for short fragments this can be an issue.

- The authors discuss multi-contact 3C+ methods, they should also include Tri-C (Hughes lab), MC-4C (de Laat lab) and Nano-C (Noordermeer lab)

- Hi-C data is often binned, but also computational methods exist that do not use binning: binless (PMID: 31028255) and shaman (https://www.biorxiv.org/content/10.1101/187203v1)

**Reviewer #2: **Abdennur and collegues present here a new python package, named pairtools, to efficiently process Hi-C based sequencing data from raw sequencing reads, to normalized contact matrices (in .cool format). Compared to other existing solutions, pairtools offers the possibility to detect multiple proximity ligation envents (called walk) which are particularly interested for long-reads sequencing technologies or to explore more complex ligation events. The python package provides a CLI which make the different steps easy to implement in a bioinformatics pipeline. The code is well packaged, easy to install and well documented. For these reasons, I think pairtools could become a valuable standard in the field of Hi-C processing in the coming years.

Overall the manuscript is well written but can be a bit difficult to follow for non experts (especially the part on the 'walk' reads). Here are a few comments on the manuscript and the tools itself ;

- The minimal pairtools-based pipeline is presented in the manuscript and is defined as mapping | parse | sort | dedup.

I think this view is misleading because it does not include any filtering step(s) to remove non valid 3C products.

The end-users could think that 90% of the sequenced reads are good 3C products while this is usully not the case.

I think the 'pairtools select' step should be part of the standard pipeline to filter non-valid 3C products (regardless how they are filtered).

- The bwa mem options are important for Hi-C data processing. I would suggest to specify which options to use in the command line exemple L102.

- After reading the manuscript, I cannot see in which cases the 'parse' command should be used instead of the 'parse2' command. It seems that 'parse2' could completly replace both commands ? if so, I would simplify the message by only presenting the 'parse2' command.

- In supFig1h, we can see that the behavior of the --report-orientation option in parse2 is different than in the parse command. What is the reason for that ? Simple recommandation or use case about which options to use would help the user.

- To better understand the interest (and the differences) of parse versus parse2, would it makes sense to always illustrate how the command will handle a single ligation and a multiple ligation. For instance, in Fig1a, I guess in a case of multiple ligation, the parse command will return a contact between the red and yellow part, while is it not reported with the parse2 command, is that correct ?

- The walk parsing strategy is not clearly explained in the Figure 2 legend. What is the message here ?

- Could you illustrate the interest of 'paritools header' with a concrete use case ? In which cases this function could be useful ?

- I always had in mind that PCR duplicates are reads that start/end exactly at the same genomic positions (and align in the same way on the genome). To my knowledge, this is the definition used by picard or samtools to remove duplicates. I'm not sure to understand, why for Hi-C data it could be interesting to account for potential losses of a few nucleotides ? what is the technical rational behind this ?

- The command 'pairtools scaling' is interesting to validate the QCs of an experiment and to define which minimal distance between two interactors could be used to remove those artefacts. However, this is not really compatible with a standard bioinformatics worflow to automatically process Hi-C data. Here again, recommandations or exemple of the 'select' command could be useful for the users.

- L297-298. The authors mentionned the interest for digestion Hi-C protocols to have access to the fraction of unwanted 3C products like self-circle, dangling-end, and mirror reads. Does pairtools provide those statistics ? or is there a way to easily get the information ?

- The comparison of the performance of the different tools is interesting but, to me, not the most important information for the end-users. Did the authors compare the performance of the tools in terms of valid ligation products detected ? is there any big differences ? This could be a way to illustrate the interest of the 'walk' reads strategy which is one of the interest of pairtools compared to the other tools. If not major difference is observed on the final list of valid ligation products between the different tools, I think this should be also mentioned.

I've tested pairtools on some MicroC data and did not had any major issue in installing it through conda and even using it in practice. I was able to get some first results in a few hours. Here are a few questions I had while running the CLI ;

- The package itself is very flexible, even maybe too flexible for beginners. I would appreciate to have some clear guidelines about what would be a typical or standard worflows for the most common Hi-C protocols, and which parameters to use for each CLI step.

- Running 'pairtools stats', I saw in the statistics file some pair types I was not able to defined ;

pair_types/UU 147151119

pair_types/Uu 43246049

pair_types/uU 43414173

pair_types/uu 14042597

pair_types/RU 1341742

pair_types/UR 1346450

What is the difference between u and U ? is there any documentation on that ?

I also noticed that in the pairs file, I only have UU ... and no uu ?

- The 'walks-policy' option of the parse command is not easy to understand. By default, I would used the '--walks-policy 5unique' for standard Hi-C protocol (digestion Hi-C) but I'm not sure to see how this option will impact the final results ?

**Have the authors made all data and (if applicable) computational code underlying the findings in their manuscript fully available?**

Reviewer #1: Yes

Reviewer #2: Yes

PLOS authors have the option to publish the peer review history of their article (what does this mean?). If published, this will include your full peer review and any attached files.

Reviewer #1: No

Reviewer #2: No
---

## [Decision Letter · Decision Letter 1]

13 May 2024

Dear Dr. Goloborodko,

We are pleased to inform you that your manuscript 'Pairtools: from sequencing data to chromosome contacts' has been provisionally accepted for publication in PLOS Computational Biology.

Best regards,

Ferhat Ay, Ph.D

Academic Editor

PLOS Computational Biology

Jian Ma

Section Editor

PLOS Computational Biology

Reviewer's Responses to Questions

**Comments to the Authors:**

Reviewer #1: The authors have incorporated the comments. I congratulate them on a nice piece of work.

Reviewer #2: The authors addressed all my previous comments. Many thanks for your work.

**Have the authors made all data and (if applicable) computational code underlying the findings in their manuscript fully available?**

Reviewer #1: Yes

Reviewer #2: Yes

PLOS authors have the option to publish the peer review history of their article (what does this mean?). If published, this will include your full peer review and any attached files.

Reviewer #1: No

Reviewer #2: **Yes: **Nicolas Servant

---

## [Editor Report · Acceptance letter]

24 May 2024

PCOMPBIOL-D-23-01631R1 

Pairtools: from sequencing data to chromosome contacts

Dear Dr Goloborodko,

I am pleased to inform you that your manuscript has been formally accepted for publication in PLOS Computational Biology. Your manuscript is now with our production department and you will be notified of the publication date in due course.

With kind regards,

Anita Estes
